# Evaluation of Fly Ash from Co-Combustion of Paper Mill Wastes and Coal as Supplementary Cementitious Materials

**DOI:** 10.3390/ma15248931

**Published:** 2022-12-14

**Authors:** Ming-Fu Wu, Wei-Hsing Huang

**Affiliations:** Department of Civil Engineering, National Central University, Taoyuan City 320317, Taiwan

**Keywords:** co-combustion, fly ash, solid recovered fuel, paper mill waste, waste-derived fuel, supplementary cementitious materials

## Abstract

The applications of waste-derived fuel from paper mills in industrial boilers benefit the reduction of carbon emissions. However, the co-combustion of waste-derived fuel and coal causes significant changes in the characteristics of the ash and brings about the need to find possible means of the utilization of the ash produced. In this work fly, ash samples were collected from circulating fluidized bed (CFB) boilers co-combusting paper mill wastes with coal and analyzed in detail. The chemical, physical, and thermal characteristics of two different co-combustion fly ashes (CCFA) were investigated using X-ray fluorescence (XRF), X-ray diffraction (XRD), thermogravimetry (TG), and scanning electron microscope (SEM). The chemical composition of CCFA is largely affected by the fuel source type. Thermal analyses of CCFA show that the type of desulfurization system used by the boiler influences the form of sulfate present in the fly ash. The presence of calcium sulfite hemihydrate can cause a high loss in the ignition of CCFA. By comparing the physical requirements specified in the ASTM standard for coal fly ash to be used in concrete, the CCFA produced from paper mill wastes was found to show good potential as supplementary cementitious materials.

## 1. Introduction

With the promotion of carbon neutrality and environmental protection policies, the paper industry has started a net-zero emission action by using combustible wastes in the paper production process as an alternative fuel, including recyclable waste paper, pulp and paper mill sludge (PPMS), and pulper rejects (PR) [1,2,3,4]. This improves the utilization efficiency of waste resources and reduces the usage of fossil fuels, consequently achieving the dual goals of sustainable development and carbon reduction. However, while the co-combustion of waste-derived fuel with coal in industrial boilers helps reduce fossil fuel consumption, the co-combustion fly ash (CCFA) experienced extensive alterations in terms of its chemical composition and material properties [4,5,6,7]. Variations in the types and sources of waste-derived fuel and the percentage of co-combustion resulted in the diversified chemical composition of the CCFA., which in turn presents difficulties in its recycling and utilization. Moreover, the existing recycling technologies and material quality standards do not apply to CCFA produced by waste-derived fuel, making their recovery and utilization difficult [5,8,9].

In general, industrial boiler type, combustion conditions, the nature of added fuels in the case of co-combustion, and process additives can significantly affect the properties of the resulting ashes. Moreover, coal or waste-derived fuels usually release sulfur oxides when burned and may cause air pollution, thus requiring strict restrictions on emissions from boiler facilities [10,11], especially circulating fluidized bed (CFB) boilers. The characteristics of the combustion ash produced by CFB boilers are significantly different from those of pulverized coal fly ash (PCFA). In addition to the relatively low combustion temperature in CFB boilers [12], in situ desulfurization and flue gas desulfurization (FGD) systems are widely used in CFB boilers to reduce sulfur dioxide emissions [13,14,15,16]. However, the captured sulfur oxides will remain in the ash and affect the characteristics of the fly ash significantly [17]. This partly resulted in the exclusion of CFB fly ash from the scope of the general standard specifications for pulverized coal fly ash (PCFA) for use in concrete [5,18,19,20].

Co-combustion of coal with alternative fuel, such as waste-derived fuel, is a relatively new development for industrial boilers. Effective utilization of the ashes produced from the co-combustion will encourage industrial boilers to adopt waste-derived fuels for co-combustion with coal and thus reducing carbon emissions. This study aimed to address the utilization of CCFAs by analyzing the chemical, physical, and mineralogical characteristics of CCFAs produced from paper mill wastes and evaluating the potential of CCFAs when used as cementitious materials in partial replacement of cement. In this study, two CCFAs were obtained from two different CFB boilers in a paper recycling mill, one was produced from the co-combustion of coal and PPMS, and the other from the co-combustion of coal and pulper rejects.

## 2. Materials and Experimental Methods

### 2.1. Waste-Derived Fuel from Paper Mill

The paper production process in a paper recycling mill (Figure 1) produces multiple types of waste, including recyclable waste paper, pulp and paper mill sludge (PPMS), and pulper rejects (PR) [21,22]. The PPMS comes from wastewater treatment sludge with main constituents of fiber, mineral fillers, and organic matter. The PPMS is converted to alternative fuel via a pressurized dehydration process. PR material is the lightweight residue produced from waste paper and paper containers treated with a hydra pulper [2,14]. The PR consists of different fibers and plastic fragments and is converted to solid recovered fuel (SRF) to be co-combusted with coal [4,21,22,23,24].

### 2.2. Co-Combustion Fly Ash from CFB Boilers

Two CCFAs from CFB boilers co-combusting two different auxiliary fuels with coal were studied in this paper. Two samples of co-combustion fly ash (CCFA) were collected separately from the storage hopper of the air pollution control unit in paper recycling mills. The first CCFA is called PPMS-CCFA and is produced from burning 60% coal and 40% PPMS, while the second is PR-CCFA, produced from burning 60% coal and 40% PR.

### 2.3. Research Methods

ASTM C1709 [25] standard guide was used as a reference for material testing in this study. In addition to testing the basic properties, various tests were used to assess the potential of CCFA as supplementary cementitious materials.

#### 2.3.1. CCFA Characteristics 

A chemical composition analysis with XRF and XRD was performed on CCFA to determine its chemical and crystalline constituents. The loss on ignition (LOI) was determined for CCFA at 550 °C and 750 °C. A laser diffraction particle size analysis was used to determine the particle size distribution of CCFA. The appearance of the fly ash was examined by using a Scanning Electron Microscope (Hitachi, FlexSEM 1000 II, Taoyuan, Taiwan).

#### 2.3.2. Evaluation as Cementitious Materials

To evaluate the suitability of CCFA as supplementary cementitious materials for use in concrete, tests were conducted for comparison with the physical and chemical requirements specified in ASTM C618 [26]. These tests included Strength Activity Index (SAI), water requirement, and soundness of cement paste or mortar prepared by replacing 20% cement with CCFA. In addition, the time of setting of cement paste was determined in accordance with ASTM C191 [27].

## 3. Results

### 3.1. CCFA Characteristics 

The chemical composition of CCFAs was determined using XRF techniques, and the results are shown in Table 1. The chemical composition indicates that the main components of PPMS-CCFA are oxygen compounds of calcium and silicon. The supplement of calcium carbonate in the paper production process provides the source of these components. The excess calcium carbonate enters the wastewater treatment system in the pulper paper production process to form PPMS, causing a high calcium carbonate content in PPMS [2,28]. Because the PPMS is rich in calcium carbonate, it is decomposed into calcium oxide and carbon dioxide during the combustion process in the boiler and undergoes an in-situ desulfurization reaction in the CFB boiler combustion chamber. As a result, the desulfurization reaction product (calcium sulfate) enters the dust collection equipment and is mixed into the fly ash [18]. While the CFB boiler using PPMS as fuel also adopts a semi-dry flue gas desulfurization plant, the use of a desulfurizer caused the PPMS-CCFA to contain calcium hydroxide (CH) and calcium sulfite hemihydrate [19,29].

PR-CCFA consists primarily of oxides of silicon, aluminum, and calcium. Unlike PPMS having a rich calcium carbonate content, PR is composed of plastic fragments and some fibers. The plastic fragments mainly come from the polyethylene (PE) or polypropylene (PP) film on paper containers [2,21]. Therefore, PR-CCFA’s chemical composition is mainly silicon and aluminum minerals. Because the CFB boiler burning pulper rejects as fuel adopts wet flue gas desulfurization, the SO_x_ in the flue gas is discharged along with the FGD wastewater, leading to a low sulfate composition in PR-CCFA.

As shown in Figure 2, the *crystalline composition* was identified using an X-ray diffraction analyzer (XRD). The results of the diffraction patterns obtained from PPMS-CCFA and PR-CCFA indicate that the fuel type has a significant impact on CCFA properties. If the waste-derived fuel contains a large amount of calcium carbonate or calcium oxide, the fuel can achieve an in situ desulfurization effect in the CFB boiler. However, the FGD gypsum is discharged to the wastewater treatment system in wet flue gas desulfurization. If the CFB boiler uses in situ desulfurization or semi-dry flue gas desulfurization, the calcium sulfite hemihydrate stays in the fly ash. Excessively high contents of calcium sulfite hemihydrates can cause an abnormal setting time when used as cementitious materials in blended cement [16,30].

Table 2 summarizes the test results on the physical properties of CCFAs. The specific gravity test result shows that PPMS-CCFA’s specific gravity is greater than that of PCFA and PR-CCFA. This is mainly due to PPMS-CCFA’s high calcium content. The particle size of fly ashes is affected by the design parameters of the dust-collecting equipment of the boiler. The fineness of the fly ash plays an important role in the performance of supplementary cementitious materials in concrete [25]. The fineness and laser particle size analysis of the fly ashes show that the particle sizes of the three fly ash sources are not very different, with the PPMS-CCFA having the lowest amount of particles retained on the #325 sieve.

According to ASTM C618, the loss on ignition (LOI) test of PCFA is conducted at a test temperature of 750 °C. Since PPMS-CCFA contains a large amount of calcium carbonate, which has a decomposition temperature of 600–800 °C [6,29,31,32], a complementary LOI test temperature of 550 °C was added to avoid the impact of weight loss due to the thermal decomposition of calcium carbonate on LOI values.

The test results showed that the weight loss of PPMS-CCFA increased from 7.2% at 550 °C to 13.2% at 750 °C. The weight loss of PR-CCFA increased from 2.7% at 550 °C to 4.7% at 750 °C. The decomposition of carbonate and the loss of the volatile inorganic compounds explain such a difference in LOI.

The XRD analysis showed that PPMS-CCFA’s main components include calcium carbonate and calcium hydroxide. TGA heating curves in Figure 3 show that the major weight losses of PPMS-CCFA are observed at 350–450 °C and 600–900 °C. The dehydration of calcium sulfite hemihydrate in PPMS-CCFA occurred at 370–400 °C [16,29]. The thermal decomposition of calcium hydroxide (CH) was observed at 400–460 °C [19,31]. These two events resulted in a weight loss of approximately 4.7%. With a weight loss of approximately 6.9%, the decomposition of calcium carbonate occurred at 600–800 °C. The TGA curve (Figure 3b) shows that the presence of calcium hydroxide and calcium carbonate leads to the difference in loss on ignition of PPMS-CCFA at different temperatures and the high amount of loss on ignition. Because PPMS-CCFA contains calcium sulfite and the decomposition reaction stage of calcium sulfite mainly occurs at 800–950 °C [16,19,33,34], calcium sulfite may change to calcium oxide at these temperatures, resulting in weight loss. The major weight loss of PR-CCFA was observed at 600–800 °C (Figure 3c). XRD analysis showed that PR-CCFA contains calcium carbonate compounds, which changed to calcium oxide, consequently causing weight loss at this temperature range [16,29].

Therefore, CCFA’s thermogravimetric loss is affected by multiple compounds, among which calcium hydroxide, calcium carbonate, and calcium sulfite hemihydrate are usually produced by in situ desulfurization or semi-dry flue gas desulfurization in CFB boilers [33]. After being collected by the electrostatic dust collection equipment and mixed into the fly ash, these compounds result in large amounts of loss on ignition of PPMS-CCFA and PR-CCFA.

SEM has been widely used to characterize fly ash’s physical or chemical properties. In this study, the SEM technique was used to examine the appearance of the samples of PCFA and CCFA produced by CFB boilers. Figure 4 shows a series of PCFA and CCFA appearance micrographs. The morphology of fly ash particles is controlled by combustion temperature and cooling rate. Because PCFA is formed after cooling in a high-temperature environment, it mainly exhibits a hollow spherical shape with a smooth surface. In contrast, the grains of CCFA have a rough surface texture and irregular shape. This is because in the operating temperature range of 850–900 °C for CFB boilers, the mineral substances in the waste-derive fuel were not subject to partial melting. The dominant particles of CCFA are granular, irregularly shaped, and mingled with some popcorn-like particles. The rough and loose surface texture of CCFA particles generates a large specific surface area that affects the water requirement adversely when used as cementitious materials [5,8,35].

Figure 4b,c shows the microscopic images of PPMS-CCFA and PR-CCFA. Due to the presence of calcium hydroxide (CH) and light calcium carbonate, the appearance of the PPMS-CCFA mainly presents typical cubic crystals and spindle-shaped crystals. However, due to different operating temperatures in the boiler, irregular particles and rough appearances can be observed in the PR-CCFA’s microcosmic images.

### 3.2. Performance Evaluation as Cementitious Materials

#### 3.2.1. Performance Testing of Cement Paste

The water consumption required for achieving a cement paste consistency and the setting time is often used to indicate the performance of fly ash used as supplementary cementitious materials. The mixed proportion of cement pastes tested for water requirement, time of setting, and autoclave expansion is shown in Table 3, along with the test results. With a replacement level of 20% of the mass of cement by CCFA, the water requirement exhibited increases of 3–5% relative to the control mixture, which is due to the surface roughness and irregular grain shape of CCFA. The initial and final setting times of the PPMS-CCFA were significantly delayed by 115 and 150 min, respectively, compared to those of the control. The initial and final setting times for PR-CCFA are similar to those of PCFA. The results show that the addition of PPMS-CCFA can seriously delay the setting time of cement paste. This is due to the presence of calcium sulfite components in PPMS-CCFA [36,37]. The increased water requirement also causes delays in the time of setting [16,30,36,37,38].

Several studies have confirmed that the presence of calcium sulfate compounds can significantly affect the hydration of cement and setting time [30,33,37,39]. Calcium sulfite has the most significant impact on the setting time of cement due to the low solubility of calcium sulfite hemihydrate in water [30,40]. When the soluble sulfate reacts with C_3_A in the cement, calcium monosulfite-aluminate (hemihydrite) is formed [16,33]. If calcium hydroxide exists simultaneously in the environment, the crystallization form of the reaction product will be affected. It covers the cement particles’ surface to take the gel form and prevents the entry of water. As a result, it prevents the C_3_A from reacting and forming calcium aluminate hydrate, leading to a prolonged setting time [16,33,39]. Compared to the setting retardation effect of calcium sulfite on cement, calcium sulfate dissolves rapidly in water. Therefore sufficient soluble sulfate can react with C_3_A in the cement to form ettringite (AFt) and fill the pores of the cement paste [16,30,41,42]. In other words, the setting retardation effect of different types of calcium sulfate on cement is different and particularly dependent upon different characteristics of the solubility and dissolution rate of each calcium sulfate [30].

The autoclave expansion of the cement paste prepared using 20% CCFAs in replacement of cement was determined after exposure to a saturated steam pressure of 20.8 ± 0.7 kgf/cm^2^ and a temperature of 215.7 ± 1.7 °C. The results are all much below the 0.8% specified by ASTM C618.

#### 3.2.2. Compressive Strength Test of Cement Mortar

According to the strength activity index test method in the ASTM C311/C311M [43], cubic mortar samples with a cube side length of 5 cm were prepared. Table 4 shows the proportions of sand and water in the cement mortar mixed with CCFAs. Under the controlled condition of a fixed fluidity value, it was found that the use of the two types of CCFA increased the water requirement to 111% and 109%. The difference in the water requirement may be caused by the irregular grain appearance and surface roughness of CCFA particles.

Figure 5 shows the variation in the compressive strength of different mortar samples. To evaluate whether CCFA can provide the pozzolanic activity or supplementary cementitious activity, the test was conducted to provide the compressive strength and the corresponding Strength Activity Indices (SAI) of the mortar samples from 3 to 90 days.

The results show that the mortar mixed with PCFA had a lower early compressive strength than the OPC. Additionally, it was found that the PCFA mortar’s late compressive strength was greater than that of the OPC in the duration between 56 and 90 days under the pozzolanic activity. According to ASTM C618, the strength activity indices of PCFA or natural pozzolanic material should be greater than 75% at 7 and 28 days. However, because CCFA is not included in the standard category of supplementary cementitious materials, reference values according to ASTM C618 can be used to judge the potential of CCFA as supplementary cementitious materials for partial replacement of cement in making concrete [25].

The test results show that PPMS-CCFA had a higher initial compressive strength than other fly ashes at three days due to the presence of calcium hydroxide (CH) in the PPMS-CCFA sample. As the age of specimens increases, the compressive strength becomes flat in the growth trend and reaches the 75% requirement at 28 days.

Although the chemical composition of PR-CCFA is similar to that of PCFA, due to the low operating temperature of the CFB boiler, the glassy phase of the material is low, inevitably resulting in a deteriorated cementitious activity. As expected, when PR-CCFA is mixed with cement, a low strength activity index of 68% was obtained, indicating that PR-CCFA has a relatively low strength activity when used in cement mixtures. However, it should be noted that the strength activity index of PR-CCFA is still increasing in the later age up to 90 days, indicating a slow pozzolanic reaction.

## 4. Conclusions

This study characterized the material properties of CCFA produced from CFB boilers in a paper recycling mill, and the potential of the CCFA as cementitious materials was evaluated by comparing the test results with the standard specifications for PCFA. The main conclusions are summarized as follows.

The material characteristics of CCFA are largely affected by the fuel source type. The high calcium content of PPMS-CCFA comes from the calcium carbonate filler in PPMS, while the chemical compositions of PR-CCFA are mainly silicon and aluminum minerals, as the fuel sources are mainly plastics and fibers.The mineralogical and thermal analyses of CCFA highlighted the differences in the type of desulfurization system in the form of sulfate present in the fly ash. The presence of calcium sulfite hemihydrate in PPMS-CCFA causes a high loss on ignition and significant increases in setting time when the ash is used in replacement of cement.The particle size of CCFA falls in the range of about 4 to 100 μm. The dominant particles of CCFA are granular and irregularly shaped, with a rough and porous surface texture. The porous surface of CCFA particles tends to increase the water requirement as the ash is used in cement mixtures.The compressive strength of mortars containing 20% CCFA shows a relatively slow development with curing time. The strength activity index of the two CCFAs studied is either satisfactory according to the requirement for coal fly ash or increasing with time and exhibiting a characteristic of pozzolanic reaction. These results confirmed that fly ash from the co-combustion of paper mill wastes and coal has the potential to be used as cementitious materials in partial replacement of cement.

This study characterized the fly ash produced from CFB boilers co-combusting paper mill wastes and coal and assessed possible means for utilization of CCFA of such origin. Further studies on the CCFA obtained from different fuel sources are highly recommended, such that feasible utilization of CCFA can be developed to facilitate the use of waste-derived fuels in industrial boilers.

## Figures and Tables

**Figure 1 materials-15-08931-f001:**
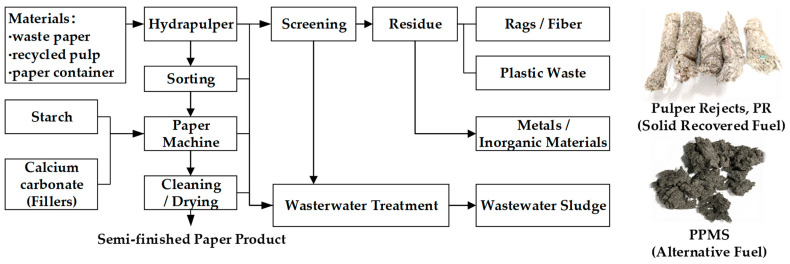
Waste-derived fuel generated during papermaking in a paper recycling mill.

**Figure 2 materials-15-08931-f002:**
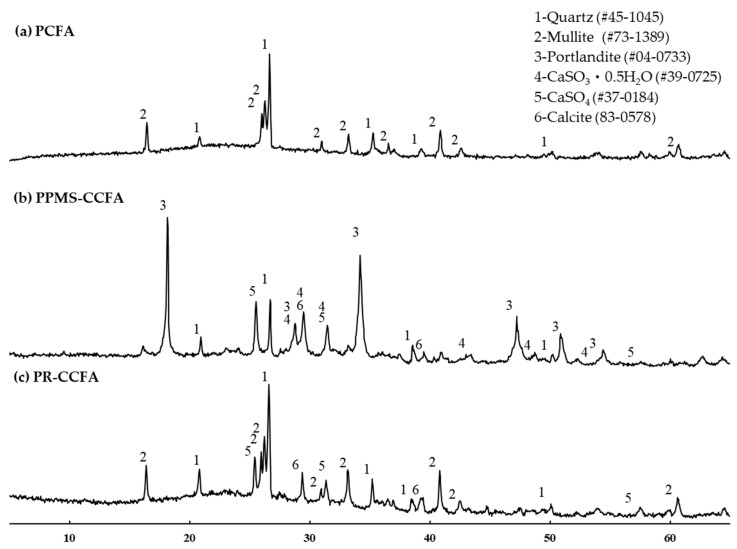
Fly ash X-Ray diffraction patterns.

**Figure 3 materials-15-08931-f003:**
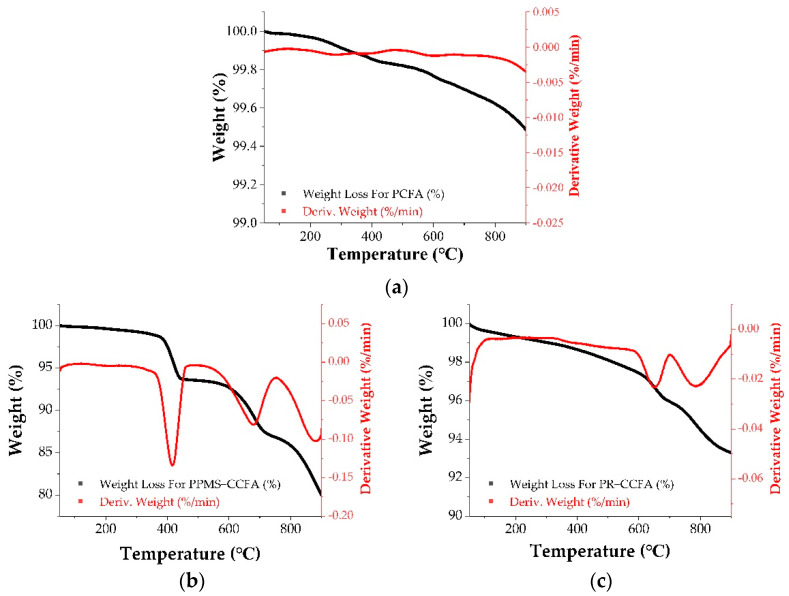
TGA and DTG heating curves of (**a**) PCFA, (**b**) PPMS-CCFA, (**c**) PR-CCFA.

**Figure 4 materials-15-08931-f004:**
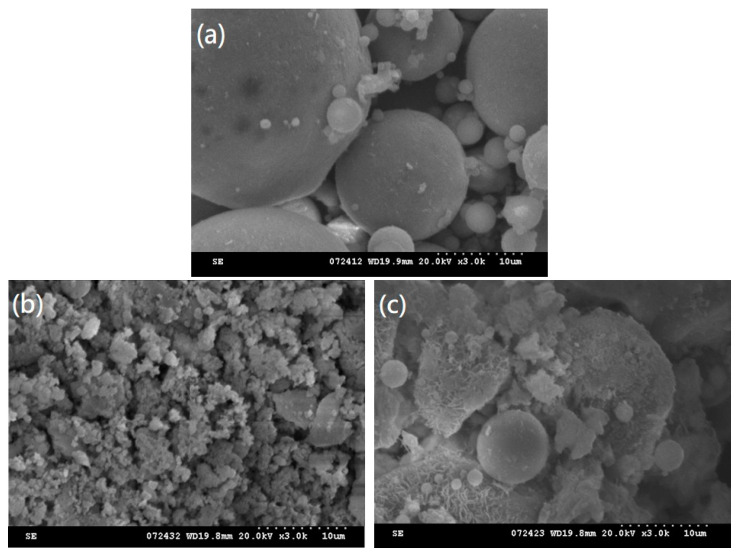
SEM micrographs of (**a**) PCFA, (**b**) PPMS-CCFA, (**c**) PR-CCFA.

**Figure 5 materials-15-08931-f005:**
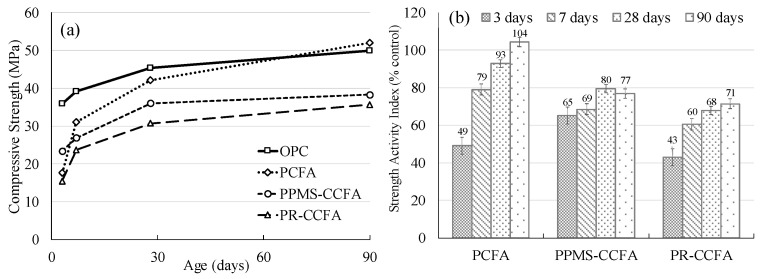
Strength development and strength activity index of different mortar mixtures (**a**) Compressive Strength, (**b**) Strength Activity Index.

**Table 1 materials-15-08931-t001:** Chemical composition of fly ashes by XRF analysis.

Sample	Chemical Composition (%)							
	SiO_2_	Al_2_O_3_	Fe_2_O_3_	CaO	MgO	TiO_2_	SO_3_	P_2_O_5_
PCFA	55.7	29.3	5.42	3.86	1.34	0.63	0.57	0.51
PPMS-CCFA	29.8	14.5	3.93	43.1	2.34	1.04	3.93	0.59
PR-CCFA	50.0	29.2	6.06	9.47	0.93	2.19	1.38	0.25

**Table 2 materials-15-08931-t002:** Physical Properties of PCFA and CCFA.

	PCFA	PPMS-CCFA	PR-CCFA
Relative Density	2.27	2.48	2.30
Fineness (retained on #325 sieve), %	20	18	24
Particle Size, µm (d_10_)	3.35	3.67	4.05
(d_50_)	21.08	21.04	26.12
(d_90_)	72.81	79.62	101.48
LOI (550 °C), %	0.3	7.2	2.7
LOI (750 °C), %	0.4	13.2	4.7
pH	10.9	8.3	11.2

**Table 3 materials-15-08931-t003:** Water demands and setting time of fly ash substituted cement paste.

Mixture ID	Mix Proportion (%)		Water Requirement for Normal Consistency (%)	Time of Setting (Minute)		Autoclave Expansion (%)
	Cement	FA		Initial Setting	Final Setting	
OPC (control)	100	0	25	165	220	0.10
PCFA	80	20	26	185	235	0.04
PPMS-CCFA	80	20	30	280	375	0.07
PR-CCFA	80	20	28	205	245	0.08

**Table 4 materials-15-08931-t004:** Mixture proportions of cement mortars.

Mixture ID	Weight (g)				Water Requirement (%)
	Cement	FA	Sand	Water	
OPC (control)	1000	0	2750	485	100
PCFA	800	200	2750	485	102
PPMS-CCFA	800	200	2750	540	111
PR-CCFA	800	200	2750	530	109

## Data Availability

Not applicable.

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
