# Peer review of "Evaluation of Fly Ash from Co-Combustion of Paper Mill Wastes and Coal as Supplementary Cementitious Materials"

_materials, 2022, doi:10.3390/ma15248931_

Round 1
Reviewer 1 Report
Herein, an experimental comparative analysis between PCFA and CCFA, investigating the impacts of different combustion process variables and fuel types on the properties of CCFA.
The work seems interesting. However, there are some issues as stated below:
1) Introduction: Should have done better paragraphing to clearly show the transition. Highlight the novelty of this study in the last paragraph.
2) Abstract: The main objective and the importance of the work should be emphasized in the abstract. In its current form, it doesn't reflect the main idea.
3) Language: Some language errors (tenses, singular/plural) and incomplete sentences in the script. Please check the unit spacing, sentence structure, tenses, and language carefully in the revised manuscript.
4) Figures and tables: Improve the figure clarity, it is blurred. Also, if it is applicable, it would be better to provide more figures to enhance its visualization.
6) Address the major limitations of this study and the possible routes to improve the limitations.
7) Cite some recent and related works from recent literature in the revised version.
8) The conclusion should be expanded with the importance of the work and the future perspectives of the work.
9) Provide XRD JCPDS numbers and discuss the purity of all chemicals.
10) Discussion of physicochemical characterizations is so raw. Please discuss it more detail by citing other works in literature with the scientific basements.Especially more discussion is needed for SEM images.
11) Discussion of XRD spectra is so raw. Please calculate d spacing of the nanomaterials and correlate it with your synthesized nanostructure properties.
Reviewer 2 Report
The study presented deals with utilization of paper waste for its applications as cement materials as blend. After careful revision, the manuscript is lacking in various areas as mentioned below.
1. The title of work is not of high clarity and could confuse the reader
2. Abstract must contain the method used and significant results
3. Introduction needs to be refined and novelty aspect must be highlighted
4. Materials section should be presented separately and must contain the information of all materials used and their pre treatment.
5. The serious concern is about the methodology. Although, the flow sheet has been presented but its needs to be improved for providing clear concepts.
6. How this fly ash was obtained ? There is no enough details regarding preparation and further its experimentation details as blend to be used?
7. what does it meant by fuel derived? either fuel was firstly derived from the material and then its spent used as fly ash? if so then details of fuel deriving concept must be presented as well.
8. The experiments must be controlled. However, the conditions and operating mechanism is lacking.
9. How many blends and in which ratio were prepared to be used as cement material.
10. CHNS analysis is important to be determined but missing
11. The bonding chemistry and mechanism as blend should be evaluated.
12. The results needs to be re-arranged with clear description and presentation quality needs to be improved.
13. Conclusion needs to be improved by providing featured outcomes and further scope.
Round 2
Reviewer 1 Report
The manuscript has been well-revised. It can be accepted to be published in its current form
Reviewer 2 Report
Authors have made changes to improve the manuscript as suggested. However, still there are some ambiguities and needs to be improved. Specific comments are as following
1. Please revise the keywords. As some Keywords are long.
2. Point 9 (Response) We have added this section to the manuscript at lines 91~96. But this line numbers is not available.
3. L129-130. The first CCFA is called PPMS-CCFA and is produced from burning 60% coal and 40% PPMS, while the second is PR-CCFA produced from burning 60% coal and 40% PR. Please discuss that why only 60/40 ratio was used. However, it could be varied to optimize the condition via suitable experimental design.
4. Section 3.2.1. Please support with reference
5. L198-199. To evaluate the suitability of CCFA as supplementary cementitious materials for use as cementitious materials. Please revise this sentence.
6. It would be more supporting by comparing the results of XRF of this study with other studies of same material by presenting in tabulated form. The same could be done for Table 2 results as well.
7. Fig. 3 I think it is Weigh loss (%) on Y axis. Please correct. Moreover improve the quality of figure.
8. Please add error difference in Fig. 5 (b)
7.
Round 3
Reviewer 2 Report
The authors have provided the details of queries and improved the manuscript.